# MetaAvatar: Learning Animatable Clothed Human Models from Few Depth Images

**Shaofei Wang**[1]
shaofei.wang@inf.ethz.ch

**Marko Mihajlovic**[1]
marko.mihajlovic@inf.ethz.ch

**Qianli Ma**[1,2]
qianli.ma@tue.mpg.de

**Andreas Geiger**[2,3]
a.geiger@uni-tuebingen.de

**Siyu Tang**[1]
siyu.tang@inf.ethz.ch

[1]ETH Zürich    [2]Max Planck Institute for Intelligent Systems, Tübingen    [3]University of Tübingen

## Abstract

In this paper, we aim to create generalizable and controllable neural signed distance fields (SDFs) that represent clothed humans from monocular depth observations. Recent advances in deep learning, especially neural implicit representations, have enabled human shape reconstruction and controllable avatar generation from different sensor inputs. However, to generate realistic cloth deformations from novel input poses, watertight meshes or dense full-body scans are usually needed as inputs. Furthermore, due to the difficulty of effectively modeling pose-dependent cloth deformations for diverse body shapes and cloth types, existing approaches resort to per-subject/cloth-type optimization from scratch, which is computationally expensive. In contrast, we propose an approach that can quickly generate realistic clothed human avatars, represented as controllable neural SDFs, given only monocular depth images. We achieve this by using meta-learning to learn an initialization of a hypernetwork that predicts the parameters of neural SDFs. The hypernetwork is conditioned on human poses and represents a clothed neural avatar that deforms non-rigidly according to the input poses. Meanwhile, it is meta-learned to effectively incorporate priors of diverse body shapes and cloth types and thus can be much faster to fine-tune, compared to models trained from scratch. We qualitatively and quantitatively show that our approach outperforms state-of-the-art approaches that require complete meshes as inputs while our approach requires only depth frames as inputs and runs orders of magnitudes faster. Furthermore, we demonstrate that our meta-learned hypernetwork is very robust, being the first to generate avatars with realistic dynamic cloth deformations given as few as 8 monocular depth frames.

## 1    Introduction

Representing clothed humans as neural implicit functions is a rising research topic in the computer vision community. Earlier works in this direction address geometric reconstruction of clothed humans from static monocular images [35, 36, 63, 64], RGBD videos [37, 38, 71, 78, 80] or sparse point clouds [12] as direct extensions of neural implicit functions for rigid objects [11, 45, 46, 52]. More recent works advocate to learn shapes in a canonical pose [7, 27, 75] in order to not only handle

35th Conference on Neural Information Processing Systems (NeurIPS 2021).

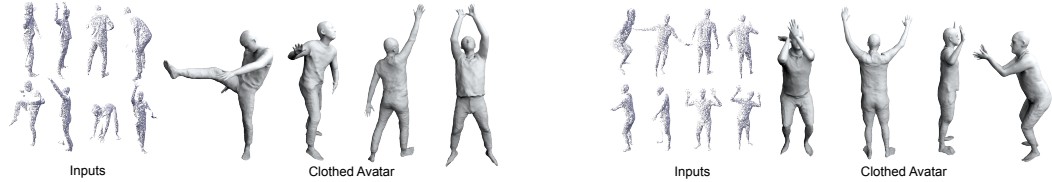

Figure 1: Given as few as 8 monocular depth images and their SMPL fittings, our meta-learned model yields a controllable neural SDF in 2 minutes which synthesizes realistic cloth deformations for unseen body poses. Here we show results of two different subjects wearing different clothes.

reconstruction, but also build controllable neural avatars from sensor inputs. However, these works do not model pose-dependent cloth deformation, limiting their realism.

On the other hand, traditional parametric human body models [41, 50, 54, 77] can represent pose-dependent soft tissue deformations of minimally-clothed human bodies. Several recent methods [13, 48] proposed to learn neural implicit functions to approximate such parametric models from watertight meshes. However, they cannot be straightforwardly extended to model clothed humans. SCANimate [65] proposed to learn canonicalized dynamic neural Signed Distance Fields (SDFs) controlled by human pose inputs and trained with Implicit Geometric Regularization (IGR [21]), thus circumventing the requirement of watertight meshes. However, SCANimate works only on dense full-body scans with accurate surface normals and further requires expensive per-subject/cloth-type training. These factors limit the applicability of SCANimate for building personalized human avatars from commodity RGBD sensors.

Contrary to all the aforementioned works, we propose to use meta-learning to effectively incorporate priors of dynamic neural SDFs of clothed humans, thus enabling fast fine-tuning (few minutes) for generating new avatars given only a few monocular depth images of unseen clothed humans as inputs. More specifically, we build upon recently proposed ideas of meta-learned initialization for implicit representations [67, 72] to enable fast fine-tuning. Similar to [67], we represent a specific category of objects (in our case, clothed human bodies in the canonical pose) with a neural implicit function and use meta-learning algorithms such as [16, 49] to learn a meta-model. However, unlike [67, 72], where the implicit functions are designed for static reconstruction, we target the generation of dynamic neural SDFs that are *controllable* by user-specified body poses. We observe that directly conditioning neural implicit functions (represented as a multi-layer perceptron) on body poses lacks the expressiveness to capture high-frequency details of diverse cloth types, and hence propose to meta-learn a hypernetwork [25] that predicts the parameters of the neural implicit function. Overall, the proposed approach, which we name *MetaAvatar*, yields controllable neural SDFs with dynamic surfaces in minutes via fast fine-tuning, given only a few depth observations of an unseen clothed human and the underlying SMPL [41] fittings (Fig. 1) as inputs. Code and data are public at https://neuralbodies.github.io/metavatar/.

## 2 Related Work

Our approach lies at the intersection of clothed human body modeling, neural implicit representations, and meta-learning. We review related works in the following.

**Clothed Human Body Modeling:** Earlier works for clothed human body modeling utilize parametric human body models [5, 26, 28, 41, 50, 54, 77] combined with deformation layers [2, 3, 7, 8] to model cloth deformations. However, these approaches cannot model fine clothing details due to their fixed topology, and they cannot handle pose-dependent cloth deformations. Mesh-based approaches that handle articulated deformations of clothes either require accurate surface registration [33, 43, 79, 83] or synthetic data [22, 24, 53] for training. Such requirement for data can be freed by using neural implicit surfaces [10, 51, 65, 73]. For example, SCANimate [65] proposed a weakly supervised approach to learn dynamic clothed human body models from 3D full-body scans which only requires minimally-clothed body registration. However, its training process usually takes one day for each subject/cloth-type combination and requires accurate surface normal information extracted from dense scans. Recent explicit clothed human models [9, 42, 44, 81] can also be learned from unregistered data. Like our method, concurrent work [44] also models pose-dependent shapes across different

subjects/cloth-types, but it requires full-body scans for training. In contrast, our approach enables learning of clothed body models in minutes from as few as 8 depth images.

**Neural Implicit Representations:** Neural implicit representations [11, 45, 46, 52, 55] have been used to tackle both image-based [27, 35, 36, 57, 59, 63, 64, 87] and point cloud-based [7, 12] clothed human reconstruction. Among these works, ARCH [27] was the first one to represent clothed human bodies as a neural implicit function in a canonical pose. However, ARCH does not handle pose-dependent cloth deformations. Most recently, SCANimate [65] proposed to condition neural implicit functions on joint-rotation vectors (in the form of unit quaternions), such that the canonicalized shapes of the neural avatars change according to the joint angles of the human body, thus representing pose-dependent cloth deformations. However, diverse and complex cloth deformations make it hard to learn a unified prior from different body shapes and cloth types, thus SCANimate resorts to per-subject/cloth-type training which is computationally expensive.

**Meta-Learning:** Meta-learning is typically used to address few-shot learning, where a few training examples of a new task are given, and the model is required to learn from these examples to achieve good performance on the new task [1, 14, 15, 17–19, 23, 29, 32, 58, 60, 62, 66, 70, 74, 76, 82, 86]. We focus on optimization-based meta-learning, where Model-Agnostic Meta Learning (MAML [16]), Reptile [49] and related alternatives are typically used to learn such models [4, 6, 20, 34, 39, 61]. In general, this line of algorithms tries to learn a "meta-model" that can be updated quickly from new observations with only few gradient steps. Recently, meta-learning has been used to learn a universal initialization of implicit representations for static neural SDFs [67] and radiance fields [72]. MetaSDF [67] demonstrates that only a few gradient update steps are needed to achieve comparable or better results than slower auto-decoder-based approaches [52]. However, [67, 72] only meta-learn static representations, whereas we are interested in dynamic representations conditioned on human body poses. To our best knowledge, we are the first to meta-learn the hypernetwork to generate the parameters of neural SDF networks.

## 3 Fundamentals

We start by briefly reviewing the linear blend skinning (LBS) method [41] and the recent implicit skinning networks [48, 65] that learn to predict skinning weights of cloth surfaces in a weakly supervised manner. Using the learned implicit skinning networks allows us to canonicalize meshes or depth observations of clothed humans, given only minimally-clothed human body model registrations to the meshes. Canonicalization of meshes or points is a necessary step as the dynamic neural SDFs introduced in Section 4 are modeled in canonical space.

### 3.1 Linear Blend Skinning

Linear blend skinning (LBS) is a commonly used technique to deform parametric human body models [5, 26, 41, 50, 54, 77] according to user-specified rigid bone transformations. Given a set of $N$ points in a canonical space, $\hat{\mathbf{X}} = \{\hat{\mathbf{x}}^{(i)}\}_{i=1}^{N}$, LBS takes a set of rigid bone transformations (in our case we use 23 local transformations plus one global transformation, assuming an underlying SMPL model) $\{\mathbf{B}_b\}_{b=1}^{24}$ as inputs, each $\mathbf{B}_b$ being a $4 \times 4$ rotation-translation matrix. For a 3D point $\hat{\mathbf{x}}^{(i)} \in \hat{\mathbf{X}}$ [1], a skinning weight vector is a probability simplex $\mathbf{w}^{(i)} \in [0,1]^{24}$, s.t. $\sum_{b=1}^{24} \mathbf{w}_b^{(i)} = 1$, that defines the affinity of the point $\hat{\mathbf{x}}^{(i)}$ to each of the bone transformations $\{\mathbf{B}_b\}_{b=1}^{24}$. The set of transformed points $\mathbf{X} = \{\mathbf{x}^{(i)}\}_{i=1}^{N}$ of the clothed human is related to $\hat{\mathbf{X}}$ via:

$$\mathbf{x}^{(i)} = LBS\left(\hat{\mathbf{x}}^{(i)}, \{\mathbf{B}_b\}, \mathbf{w}^{(i)}\right) = \left(\sum_{b=1}^{24} \mathbf{w}_b^{(i)} \mathbf{B}_b\right) \hat{\mathbf{x}}^{(i)}, \quad \forall i = 1, \ldots, N \tag{1}$$

$$\hat{\mathbf{x}}^{(i)} = LBS^{-1}\left(\mathbf{x}^{(i)}, \{\mathbf{B}_b\}, \mathbf{w}^{(i)}\right) = \left(\sum_{b=1}^{24} \mathbf{w}_b^{(i)} \mathbf{B}_b\right)^{-1} \mathbf{x}^{(i)}, \quad \forall i = 1, \ldots, N \tag{2}$$

---

[1] with slight abuse of notation, we also use $\hat{\mathbf{x}}$ to represent points in homogeneous coordinates when necessary.

where Eq. (1) is referred to as the LBS function and Eq. (2) is referred to as the inverse-LBS function. The process of applying Eq. (1) to all points in $\hat{\mathbf{X}}$ is often referred to as *forward skinning* while the process of applying Eq. (2) is referred to as *inverse skinning*.

## 3.2 Implicit Skinning Networks

Recent articulated implicit representations [48, 65] have proposed to learn functions that predict the forward/inverse skinning weights for arbitrary points in $\mathbb{R}^3$. We follow this approach, but take advantage of a convolutional point-cloud encoder [56] for improved generalization. Formally, we define the implicit forward and inverse skinning networks as $h_{\text{fwd}}(\cdot, \cdot) : (\mathbb{R}^{3 \times K}, \mathbb{R}^3) \mapsto \mathbb{R}^{24}$ and $h_{\text{inv}}(\cdot, \cdot) : (\mathbb{R}^{3 \times K}, \mathbb{R}^3) \mapsto \mathbb{R}^{24}$, respectively. Both networks take as input a point cloud with $K$ points and a query point for which they predict skinning weights. Therefore, we can analogously re-define Eq. (1, 2) respectively as:

$$\mathbf{x}^{(i)} = \left( \sum_{b=1}^{24} h_{\text{fwd}}(\hat{\mathbf{X}}, \hat{\mathbf{x}}^{(i)})_b \mathbf{B}_b \right) \hat{\mathbf{x}}^{(i)}, \quad \forall i = 1, \ldots, N \tag{3}$$

$$\hat{\mathbf{x}}^{(i)} = \left( \sum_{b=1}^{24} h_{\text{inv}}(\mathbf{X}, \mathbf{x}^{(i)})_b \mathbf{B}_b \right)^{-1} \mathbf{x}^{(i)}, \quad \forall i = 1, \ldots, N \tag{4}$$

**Training the Skinning Network:** We follow the setting of SCANimate [65], where a dataset of observed point clouds $\{\mathbf{X}\}$ and their underlying SMPL registration are known. For a sample $\mathbf{X}$ in the dataset, we first define the re-projected points $\bar{\mathbf{X}} = \{\bar{\mathbf{x}}\}_{i=1}^N$ as $\mathbf{X}$ mapped to canonical space via Eq. (4) and then mapped back to transformed space via Eq. (3). We then define the training loss:

$$\mathcal{L}(\mathbf{X}) = \lambda_r \mathcal{L}_r + \lambda_s \mathcal{L}_s + \lambda_{skin} \mathcal{L}_{skin}, \tag{5}$$

where $\mathcal{L}_r$ represents a re-projection loss that penalizes the L2 distance between an input point $\mathbf{x}$ and the re-projected point $\bar{\mathbf{x}}$, $\mathcal{L}_s$ represents L1 distances between the predicted forward skinning weights and inverse skinning weights, and $\mathcal{L}_{skin}$ represents the L1 distances between the predicted (forward and inverse) skinning weights and the barycentrically interpolated skinning weights $\mathbf{w}^{(i)}$ on the registered SMPL shape that is closest to point $\mathbf{x}^{(i)}$; please refer to the Supp. Mat. for hyperparameters and details.

We train two skinning network types, the first one takes a partial point cloud extracted from a depth image as input and performs the *inverse skinning*, while the second one takes a full point cloud sampled from iso-surface points generated from the dynamic neural SDF in the canonical space and performs *forward skinning*.

**Canonicalization:** We use the learned inverse skinning network to canonicalize complete or partial point clouds $\{\hat{\mathbf{X}}\}$ via Eq. (4) which are further used to learn the canonicalized dynamic neural SDFs.

## 4 MetaAvatar

Our approach meta-learns a unified clothing deformation prior from the training set that consists of different subjects wearing different clothes. This meta-learned model is further efficiently fine-tuned to produce a dynamic neural SDF from an arbitrary amount of fine-tuning data of unseen subjects. In extreme cases, MetaAvatar requires as few as 8 depth frames and takes only 2 minutes for fine-tuning to yield a subject/cloth-type-specific dynamic neural SDF (Fig. 1).

We assume that each subject/cloth-type combination in the training set has a set of registered bone transformations and canonicalized points, denoted as $\{\{\mathbf{B}_b\}_{b=1}^{24}, \hat{\mathbf{X}}\}$. Points in $\hat{\mathbf{X}}$ are normalized to the range $[-1, 1]^3$ according to their corresponding registered SMPL shape. With slight abuse of notation, we also define $\mathbf{X}$ as all possible points in $[-1, 1]^3$. Our goal is to meta-learn a hyper-network [25, 69] which takes $\{\mathbf{B}_b\}_{b=1}^{24}$ ($\{\mathbf{B}_b\}$ for shorthand) as inputs and predicts *parameters of the neural SDFs* in the canonical space. Denoting the hypernetwork as $g_\psi(\{\mathbf{B}_b\})$ and the predicted

neural SDF as $f_\phi(\mathbf{x})|_{\phi=g_\psi(\{\mathbf{B}_b\})}$, we use the following IGR [21] loss to supervise the learning of $g$:

$$\mathcal{L}_{\text{IGR}}(f_\phi(\hat{\mathbf{X}})|_{\phi=g_\psi(\{\mathbf{B}_b\})}) = \sum_{\mathbf{x}\in\hat{\mathbf{X}}} \lambda_{sdf}\left|f_\phi(\mathbf{x})|_{\phi=g_\psi(\{\mathbf{B}_b\})}\right| + \lambda_{\mathbf{n}}\left(1 - \langle\mathbf{n}(\mathbf{x}), \nabla_{\mathbf{x}} f_\phi(\mathbf{x})|_{\phi=g_\psi(\{\mathbf{B}_b\})}\rangle\right)$$
$$+ \lambda_E\left|\|\nabla_{\mathbf{x}} f_\phi(\mathbf{x})|_{\phi=g_\psi(\{\mathbf{B}_b\})}\|_2 - 1\right| \qquad \text{(on-surface loss)}$$
$$+ \sum_{\mathbf{x}\sim\mathbf{X}\backslash\hat{\mathbf{X}}} \lambda_O \exp\left(-\alpha\cdot\left|f_\phi(\mathbf{x})|_{\phi=g_\psi(\{\mathbf{B}_b\})}\right|\right)$$
$$+ \lambda_E\left|\|\nabla_{\mathbf{x}} f_\phi(\mathbf{x})|_{\phi=g_\psi(\{\mathbf{B}_b\})}\|_2 - 1\right| \qquad \text{(off-surface loss)}$$

$$(6)$$

where $\mathbf{n_x}$ is the surface normal of point $\mathbf{x}$. We assume that this information, along with the ground-truth correspondences from transformed space to canonical space is available when learning the meta-model on the training set, but do not require this for fine-tuning $g$ on unseen subjects.

In practice, we found that directly learning the hypernetwork $g_\psi$ via Eq. (6) does not converge, and thus we decompose the meta-learning of $g_\psi$ into two steps. First, we learn a meta-SDF [67] (without conditioning on $\{\mathbf{B}_b\}$, Sec. 4.1), and then we meta-learn a hypernetwork that takes $\{\mathbf{B}_b\}$ as input and predicts the *residuals* to the parameters of the previously learned meta-SDF (Sec. 4.2).

### 4.1 Meta-learned Initialization of Static Neural SDFs

To effectively learn a statistical prior of clothed human bodies, we ignore the input bone transformations $\{\mathbf{B}_b\}$ and meta-learn the static neural SDF $f_\phi(\mathbf{x}) : [-1,1]^3 \mapsto \mathbb{R}$, parameterized by $\phi$, from all canonicalized points of subjects *with different genders, body shapes, cloth types, and poses*. Furthermore, for faster and more stable convergence, the neural SDF $f_\phi$ function additionally leverages the periodic activation functions [68].

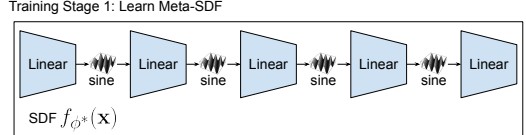

Figure 2: **Overview of the meta-SDF network.** We use a 5-layer SIREN [67] network with 256 neurons for each layer.

The full meta-learning algorithm for the static neural SDFs is described in Alg. 1.

---

**Algorithm 1** Meta-learning SDF with Reptile [49]
___

    **Initialize**: meta-network parameters $\phi$, meta learning rate $\beta$, inner learning rate $\alpha$, max training iteration $N$, inner-loop iteration $m$, batch size $M$

1: **for** $i = 1, \ldots, N$ **do**
2:     Sample a batch of $M$ training samples $\{\hat{\mathbf{X}}^{(j)}\}_{j=1}^M$
3:     **for** $j = 1, \ldots, M$ **do**
4:         $\phi_0^{(j)} = \phi$
5:         **for** $k = 1, \ldots, m$ **do**
6:             $\phi_k^{(j)} = \phi_{k-1}^{(j)} - \alpha\nabla_\phi\mathcal{L}_{\text{IGR}}(f_\phi(\hat{\mathbf{X}}^{(j)})|_{\phi=\phi_{k-1}^{(j)}})$
7:         **end for**
8:     **end for**
9:     $\phi \leftarrow \phi + \beta\frac{1}{M}\sum_{j=1}^M(\phi_m^{(j)} - \phi)$
10: **end for**

---

### 4.2 Meta-learned Initialization of HyperNetwork for Dynamic Neural SDFs

The meta-learned static neural SDF explained in the previous section can efficiently adapt to new observations, however it is not controllable by user-specified bone transformations $\{\mathbf{B}_b\}$. Therefore, to enable non-rigid pose-dependent cloth deformations, we further meta-learn a hypernetwork [25] to predict *residuals* to the learned parameters of the meta-SDF in Alg. 1.

The key motivation for meta-learning the hypernetwork is to build an effective unified prior for articulated clothed humans, which enables the recovery of the non-rigid clothing deformations at test time via the efficient fine-tuning process from several depth images of unseen subjects.

---

**Algorithm 2** Meta-learning hypernetwork with Modified Reptile

**Initialize**: meta-hypernetwork parameters $\psi$, pre-trained meta-SDF parameters $\phi^*$, meta learning rate $\beta$, inner learning rate $\alpha$, max training iteration $N$, inner-loop iteration $m$.
1: **for** $i = 1, \ldots, N$ **do**
2: $\quad \psi_0 = \psi$
3: $\quad$ Randomly choose a subject/cloth-type combination $n$
4: $\quad$ Uniformly sample $M \sim \{1, \ldots, D^{(n)}\}$ where $D^{(n)}$ is the number of datapoints of subject/cloth-type combination $n$
5: $\quad$ Sample $M$ datapoints from subject/cloth-type combination $n$, denoting these datapoints as $\mathcal{S} = \{\{\mathbf{B}_b\}^{(j)}, \hat{\mathbf{X}}^{(j)}\}_{j=1}^{M}$
6: $\quad$ **for** $k = 1, \ldots, m$ **do**
7: $\quad\quad \mathcal{L} = \frac{1}{M} \sum_{(\{\mathbf{B}_b\}, \hat{\mathbf{X}}) \in \mathcal{S}} \mathcal{L}_{\text{IGR}}(f_\phi(\hat{\mathbf{X}})|_{\phi = g_{\psi_{k-1}}(\{\mathbf{B}_b\}) + \phi^*})$
8: $\quad\quad \psi_k = \psi_{k-1} - \alpha \nabla_{\psi_{k-1}} \mathcal{L}$
9: $\quad$ **end for**
10: $\quad \psi \leftarrow \psi + \beta(\psi_m - \psi_0)$
11: **end for**

---

Denoting the meta-SDF learned by Alg. 1 as $\phi^*$ and our hypernetwork as $g_\psi(\{\mathbf{B}_b\})$, we implement Alg. 2. This algorithm differs from the original Reptile [49] algorithm in that it tries to optimize the inner-loop on arbitrary amount of data. Note that for brevity the loss in the inner-loop (line 7-line 8) is computed over the whole batch $\mathcal{S}$, whereas in practice we used stochastic gradient descent (SGD) with fixed mini-batch size over $\mathcal{S}$ since $\mathcal{S}$ can contain hundreds of samples; SGD is used with the mini-batch size of 12 for the inner-loop.

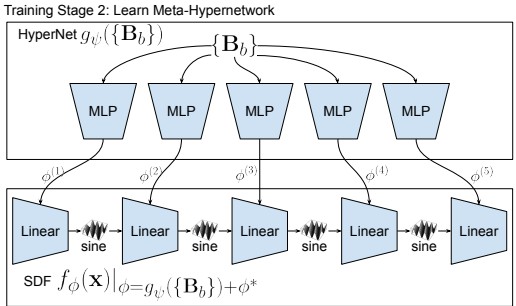

Figure 3: **Overview of the meta-hypernetwork**. It predicts residuals to $\phi^*$ which is learned in Sec. 4.1

**Inference:** At test-time, we are given a small fine-tuning set $\{\{\mathbf{B}_b\}^{fine,(j)}, \hat{\mathbf{X}}^{fine,(j)}\}_{j=1}^{M}$ and the validation set $\{\{\mathbf{B}_b\}^{val,(j)}\}_{j=1}^{K}$. The fine-tuning set is used to optimize the hypernetwork parameters $\psi$ ($m = 256$ SGD epochs) that are then used to generate neural SDFs from bone transformations available in the validation set. The overall inference pipeline including the inverse and the forward LBS stages is shown in Fig. 4.

**Bone Transformation Encoding:** We found that a small hierarchical MLP proposed in LEAP [48] for encoding bone transformations works slightly better than the encoding of unit quaternions used in SCANimate [65]. Thus, we employ the hierarchical MLP encoder to encode $\{\mathbf{B}_b\}$ for $g$ unless specified otherwise; we ablate different encoding types in the experiment section.

## 5 Experiments

We validate the proposed MetaAvatar model for learning meta-models and controllable dynamic neural SDFs of clothed humans by first comparing our MetaAvatar to the established approaches [13, 48, 65]. Then, we ablate the modeling choices for the proposed controllable neural SDFs. And lastly, we demonstrate MetaAvatar's capability to tackle the challenging task of learning animatable clothed human models from reduced data, to the point that only 8 depth images are available as input.

**Datasets:** We use the CAPE dataset [43] as the major test bed for our experiments. This dataset consists of 148584 pairs of clothed meshes, capturing 15 human subjects wearing different clothes

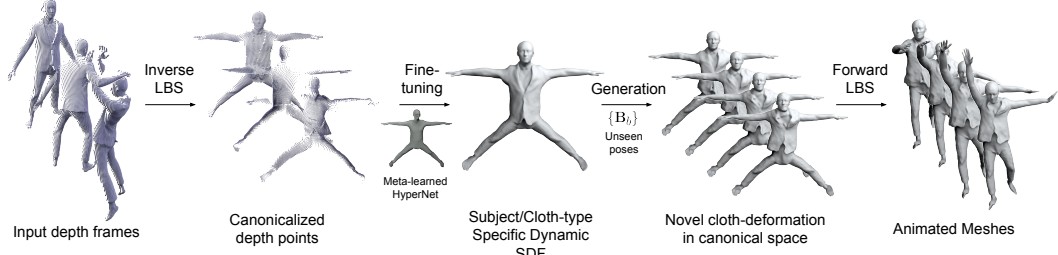

Figure 4: **Overview of our inference pipeline**. The inverse LBS net (Sec. 3.2) takes a small set of input depth frames together with their underlying SMPL registrations to canonicalize the depth points; then the meta-learned hypernetwork (Sec. 4.2) is fine-tuned to represent the instance specific dynamic SDF; given novel poses, the updated hypernetwork generates pose-dependent cloth-deformations in canonical space, and the animated meshes are obtained via the forward LBS network (Sec. 3.2).

while performing different actions. We use 10 subjects for meta-learning, which we denote as the training set. We use four unseen subjects (00122, 00134, 00215, 03375)[2] for fine-tuning and validation; for each of these four subjects, the corresponding action sequences are split into fine-tuning set and validation set. The fine-tuning set is used for fine-tuning the MetaAvatar models, it is also used to evaluate pose interpolation task. The validation set is used for evaluating novel pose extrapolation. Among the four unseen subjects, two of them (00122, 00215) perform actions that are present in the training set for the meta-learning; we randomly split actions of these two subjects with 70% fine-tuning and 30% validation. Subject 00134 and 03375 perform two trials of actions unseen in the training set for meta-learning. We use the first trial as the fine-tuning set and the second trial as the validation set. Subject 03375 also has one cloth type (blazer) that is unseen during meta-learning.

**Baselines:** We use NASA [13], LEAP [48], and SCANimate [65] as our baselines. NASA and SCANimate cannot handle multi-subject-cloth with a single model so we train per-subject/cloth-type models from scratch for each of them on the fine-tuning set. LEAP is a generalizable neural-implicit human body model that has shown to work on minimally-clothed bodies. We extend LEAP by adding a one-hot encoding to represent different cloth types (similarly to [43]) and train it jointly on the full training and the fine-tuning set.

As for the input format, we use depth frames rendered from CAPE meshes for our MetaAvatar. To render the depth frames, we fixed the camera and rotate the CAPE meshes around the y-axis (in SMPL space) at different angles with an interval of 45 degrees; note that for each mesh we only render it on one angle, simulating a monocular camera taking a round-view of a moving person. For the baselines, we use watertight meshes and provide the occupancy [45] loss to supervise the training of NASA and LEAP, while sampling surface points and normals on watertight meshes to provide the IGR loss supervision for SCANimate. Note that our model is at great disadvantage, as for fine-tuning we only use discrete monocular depth observations without accurate surface normal information.

**Tasks and Evaluation:** Our goal is to generate realistic clothing deformations from arbitrary input human pose parameters. To systematically understand the generalization capability of the MetaAvatar representation, we validate the baselines and MetaAvatar on two tasks, pose interpolation and extrapolation. For interpolation, we sample every 10th frame on the fine-tuning set for training/fine-tuning, and sample every 5th frame (excluding the training frames) also on the fine-tuning set for validation. For extrapolation, we sample every 5th frame on the fine-tuning set for training, and sample every 5th frame on the validation set for validation.

Interpolation is evaluated using three metrics: point-based ground-truth-to-mesh distance ($D_p \downarrow$, in cm), face-based ground-truth-to-mesh distance ($D_f \downarrow$, in cm), and point-based ground-truth-to-mesh normal consistency ($NC \uparrow$, in range $[-1, 1]$). For computing these interpolation metrics we ignore non-clothed body parts such as hands, feet, head, and neck. For extrapolation, we note that cloth-deformation are often stochastic; in such a case, predicting overly smooth surfaces can result in lower distances and higher normal consistency. Thus, we also conduct a large-scale perceptual study using

---

[2]We ignore subject 00159 because it has significantly less data compared to other subjects.

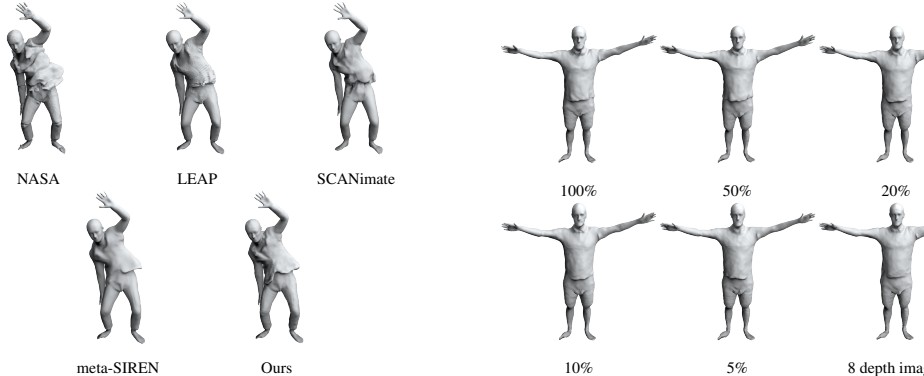

Figure 5: Qualitative comparison on extrapolation results with blazer outfit. NASA shows consistent blocky artifacts. LEAP predicts overly smooth surfaces missing the tails of the blazer outfit. SCANimate does not generalize as this specific pose has not been seen during training. Directly meta-learning a SIREN [68] network that conditions on input poses produces a smooth surface that does not capture the blazer tails well.

Figure 6: Qualitative comparison on extrapolation results when reducing fine-tuning data on subject 00215 wearing poloshirt. The caption indicates the amount of fine-tuning data used to fine-tune the meta-hypernetwork on this unseen subject. Our meta-learned model captures for this unseen subject the sliding effect of the poloshirt at this pose in which the person raising arms, even fine-tuned with just 8 depth images.

Amazon Mechanical Turk, and report the perceptual scores (PS ↑) which reflects the percentage of users who favor the outputs of baselines over MetaAvatar. Details about user study design can be found in the Supp. Mat.

## 5.1 Evaluation Against Baselines

In this section, we report results on both interpolation and extrapolation tasks against various baselines described above. For NASA and SCANimate, we train one model for each subject/cloth-type combination on the fine-tuning set. For them it usually takes several thousand of epochs to converge for each subject/cloth-type combination, which roughly equals to 10-24 hours of training. For LEAP, we train a single model on both the training and the fine-tuning set using two days. For the MetaAvatar, we meta-learn a single model on the training set, and for each subject/cloth-type combination we fine-tune the model for 256 epochs to produce subject/cloth-type specific models. The exact fine-tuning time ranges from 40 minutes to 3 hours depending on the amount of available data since we are running a fixed number of epochs; see the Supp. Mat. for detailed runtime comparison on each subject/cloth-type combination. Note that MetaAvatar uses *partial* depth observations while the other baselines are trained on *complete* meshes. The results are reported in Table 1.

Importantly, models of NASA and SCANimate are over-fitted to each subject/cloth-type combination as they cannot straightforwardly leverage prior knowledge from multiple training subjects. LEAP is trained on all training and fine-tuning data with input encodings to distinguish different body shapes and cloth types, but it fails to capture high-frequency details of clothes, often predicting smooth surfaces (Fig. 5); this is evidenced by its lower perceptual scores (PS) compared to SCANimate and our MetaAvatar. In contrast to these baselines, MetaAvatar successfully captures a unified clothing deformation prior of diverse body shapes and cloth types, which generalizes well to unseen body shapes (00122, 00215), unseen poses (00134, 03375), and unseen cloth types (03375 with blazer outfit); although we did not outperform LEAP on the interpolation task for subject 00134 and 03375, we note that 1) our method uses only 2.5D input for fine-tuning, while LEAP has access to ground-truth canonical meshes during training; 2) subject 00134 and 03375 comprise much more missing frames than subject 00122 and 00215, resulting in higher stochasticity and thus predicting smooth surfaces (such as LEAP) may yield better performance; this is also evidenced by LEAP's much lower perceptual scores on subject 00134 and 03375, although obtaining the best performance for pose interpolation. We encourage the readers to watch the side-by-side comparison videos available on our project page: https://neuralbodies.github.io/metavatar/.

|  |  | 3D Input | | | 2.5D Input |
|---|---|---|---|---|---|
|  |  | NASA | LEAP | SCANimate | Ours |
| Subj 00122, 00215 | | | | | |
| Ex. | PS $\uparrow$ | 0.078 | 0.314 | 0.333 | **0.5** |
| Int. | $D_p \downarrow$ | 0.484 | 0.454 | 0.586 | **0.450** |
|  | $D_f \downarrow$ | 0.327 | 0.293 | 0.489 | **0.273** |
|  | $NC \uparrow$ | 0.752 | 0.807 | 0.793 | **0.821** |
| Subj 00134, 03375 | | | | | |
| Ex. | PS $\uparrow$ | 0.182 | 0.224 | 0.481 | **0.5** |
| Int. | $D_p \downarrow$ | 0.595 | **0.483** | 0.629 | 0.518 |
|  | $D_f \downarrow$ | 0.469 | **0.340** | 0.542 | 0.367 |
|  | $NC \uparrow$ | 0.693 | **0.780** | 0.755 | 0.773 |
| Averge per-model training/fine-tuning time (hours) | | | | | |
|  |  | >10 | - | >10 | 1.60 |

Table 1: **Comparison to baselines.** $D_p$, $D_f$ and $NC$ are reported for interpolation (Int.) while PS is reported for extrapolation (Ex.). Note that MetaAvatar is fine-tuned on *depth images* while all other baselines are trained on *complete meshes*. The training/fine-tuning times are just rough estimates, as ours does not include the time for meta-learning, while many factors, including varying training schedules, disk-IOs and hardware setups, can affect the final speed.

|  |  | MLP | PosEnc | SIREN | Hyper Quat | Hyper BoneEnc |
|---|---|---|---|---|---|---|
| Subj 00122, 00215 | | | | | | |
| Int. | $D_p \downarrow$ | 3.278 | 1.806 | 0.472 | **0.460** | 0.461 |
|  | $D_f \downarrow$ | 2.201 | 0.998 | 0.301 | 0.288 | **0.288** |
|  | $NC \uparrow$ | -0.279 | -0.045 | 0.815 | 0.818 | **0.820** |
| Subj 00134, 03375 | | | | | | |
| Int. | $D_p \downarrow$ | 3.320 | 1.498 | 0.532 | 0.526 | **0.523** |
|  | $D_f \downarrow$ | 2.190 | 0.772 | 0.385 | 0.378 | **0.374** |
|  | $NC \uparrow$ | -0.300 | -0.099 | **0.773** | 0.772 | 0.772 |

Table 2: **Ablation for different architectures on the interpolation task.** Hyper-Quat is our model that takes the relative joint-rotations (in the form of unit quaternions) as inputs. Hyper-BoneEnc is our full model with hierarchical bone encoding MLP of LEAP [48]. Models in the table are fine-tuned for 128 epochs.

## 5.2 Ablation Study on Model Architectures

We further ablate model architecture choices for MetaAvatar. We compare against (1) a plain MLP that takes the concatenation of the relative joint-rotations (in the form of unit quaternions) and query points as input (MLP), (2) a MLP that takes the concatenation of the relative joint-rotations and the positional encodings of query point coordinates as input (PosEnc), and (3) a SIREN network that takes the concatenation of the relative joint-rotations and query points as input (SIREN). The evaluation task is interpolation; results are reported in Table 2. For the baselines (MLP, PosEnc and SIREN), we directly use Alg. 2 to meta-learn the corresponding models with $\phi^* = 0$. For MLP and PosEnc, the corresponding models fail to produce reasonable shapes. For SIREN, it produces unnaturally smooth surfaces which cannot capture fine clothing details such as wrinkles (Fig. 5).

## 5.3 Few-shot learning of MetaAvatar

In this section, we evaluate the few-shot learning capabilities of MetaAvatar. As shown in Table 3, we reduce the amount of data on the fine-tuning set, and report the performance of models fine-tuned on reduced amount of data. Note that with <1% data, we require only one frame from each action sequence available for a subject/cloth-type combination, this roughly equals to 8-20 depth frames depending on the amount of data for that subject/cloth-type combination. For interpolation, the performance drops because the stochastic nature of cloth deformation becomes dominant when the amount of fine-tuning data decreases. On the other hand, the perceptual scores (PS) are better than NASA and LEAP even with **<1%** data in the form of **partial depth observations**, and better than or comparable to SCANimate in most cases.

| Fine-tune data (%) | | 100 | 50 | 20 | 10 | 5 | <1 |
|---|---|---|---|---|---|---|---|
| Subj 00122, 00215 | | | | | | | |
| Ex. | PS $\uparrow$ | 0.5 | 0.471 | 0.509 | 0.473 | 0.373 | **0.510** |
| Int. | $D_p \downarrow$ | - | **0.450** | 0.480 | 0.512 | 0.543 | 0.592 |
|  | $D_f \downarrow$ | - | **0.273** | 0.310 | 0.353 | 0.391 | 0.450 |
|  | $NC \uparrow$ | - | **0.821** | 0.808 | 0.795 | 0.785 | 0.768 |
| Subj 00134, 03375 | | | | | | | |
| Ex. | PS $\uparrow$ | **0.5** | 0.476 | 0.424 | 0.463 | 0.439 | 0.387 |
| Int. | $D_p \downarrow$ | - | **0.518** | 0.545 | 0.576 | 0.603 | 0.619 |
|  | $D_f \downarrow$ | - | **0.367** | 0.400 | 0.438 | 0.471 | 0.489 |
|  | $NC \uparrow$ | - | **0.773** | 0.762 | 0.753 | 0.745 | 0.737 |
| Average per-model training/fine-tuning time (hours) | | | | | | | |
|  |  | 1.60 | 0.8 | 0.32 | 0.16 | 0.08 | 0.02 |

Table 3: **Ablation for few-shot learning.** We report performance of MetaAvatar on reduced amount of fine-tuning data. Fine-tuning time scales linearly with the amount of data, since we run for a fixed number of epochs.

The qualitative comparison on extrapolation results of reduced fine-tuning data is shown in Fig. 6. Please see the Supp. Mat. for more qualitative results on few-shot learning, including results on depth from raw scans, results on real depth images and comparison with pre-trained SCANimate model.

## 6 Conclusion

We introduced MetaAvatar, a meta-learned hypernetwork that represents controllable dynamic neural SDFs applicable for generating clothed human avatars. Compared to existing methods, MetaAvatar learns from less data (temporally discrete monocular depth frames) and requires less time to represent novel unseen clothed humans. We demonstrated that the meta-learned deformation prior is robust and can be used to effectively generate realistic clothed human avatars in 2 minutes from as few as 8 depth observations.

MetaAvatar is compatible with automatic registration methods [8, 75], human motion models [84, 85], implicit hand models [30, 31] and rendering primitives [40, 47] that could jointly enable an efficient end-to-end photo-realistic digitization of humans from commodity RGBD sensors, which has broad applicability in movies, games, and telepresence applications. However, this digitization may raise privacy concerns that need to be addressed carefully before deploying the introduced technology.

## 7 Acknowledgment

Siyu Tang acknowledges funding by the Swiss National Science Foundation under project 200021_204840. Andreas Geiger was supported by the ERC Starting Grant LEGO-3D (850533) and DFG EXC number 2064/1 - project number 390727645. We thank Jinlong Yang for providing results of SCANimate on the CAPE dataset. We thank Yebin Liu for sharing POSEFusion [38] data and results during the rebuttal period. We also thank Yan Zhang, Siwei Zhang and Korrawe Karunratanakul for proof reading the paper.

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
