# – Supplementary Material –
# MetaAvatar: Learning Animatable Clothed Human Models from Few Depth Images

**Shaofei Wang**[1]
shaofei.wang@inf.ethz.ch

**Marko Mihajlovic**[1]
marko.mihajlovic@inf.ethz.ch

**Qianli Ma**[1,2]
qianli.ma@tue.mpg.de

**Andreas Geiger**[2,3]
a.geiger@uni-tuebingen.de

**Siyu Tang**[1]
siyu.tang@inf.ethz.ch

[1]ETH Zürich    [2]Max Planck Institute for Intelligent Systems, Tübingen    [3]University of Tübingen

## Abstract

In this supplementary material, we first specify the exact losses and hyperparameters used for our forward and inverse skinning networks (Sec. 1). Then we describe hyperparameters for the proposed MetaAvatar, baselines (NASA, LEAP, SCANimate), and ablation baselines (MLP, PosEnc, SIREN) in Sec. 2, Sec. 3 and Sec. 4 respectively. Next, we present our user interface for conducting the perceptual study and define the perceptual score (PS) reported in the main paper (Sec. 5). We further report timing comparisons on each validation subject/cloth-type combination (Sec. 6) and additional qualitative results (Sec. 7). Lastly, we discuss the limitations of our approach (Sec. 8).

## 1 Loss Definitions and Hyperparameters for Skinning Networks

In Section 3 of the main paper, we followed [9, 14] and defined the loss for training the skinning networks as follows:

$$\mathcal{L}(\mathbf{X}) = \lambda_r \mathcal{L}_r + \lambda_s \mathcal{L}_s + \lambda_{skin} \mathcal{L}_{skin} , \tag{1}$$

where $\mathcal{L}_r$ represents a re-projection loss that penalizes the $l_2$ distance between an input point $\mathbf{x}$ and the corresponding re-projected point $\bar{\mathbf{x}}$:

$$\mathcal{L}_r(\mathbf{X}) = \frac{1}{N} \sum_{i=1}^{N} \|\mathbf{x}^{(i)} - \bar{\mathbf{x}}^{(i)}\|_2 , \tag{2}$$

$\mathcal{L}_s$ is the mean $l_1$ distances between the predicted forward skinning weights and inverse skinning weights:

$$\mathcal{L}_s(\mathbf{X}) = \frac{1}{N} \sum_{i=1}^{N} \sum_{b=1}^{24} |h_{\text{fwd}}(\hat{\mathbf{X}}, \hat{\mathbf{x}}^{(i)})_b - h_{\text{inv}}(\mathbf{X}, \mathbf{x}^{(i)})_b| , \tag{3}$$

and $\mathcal{L}_{skin}$ represents the mean $l_1$ distance between the predicted (forward and inverse) skinning weights and the barycentrically interpolated skinning weights $\mathbf{w}^{(i)}$ on the registered minimally-

35th Conference on Neural Information Processing Systems (NeurIPS 2021).

clothed shape that is closest to point $\mathbf{x}^{(i)}$:

$$\mathcal{L}_{skin}(\mathbf{X}) = \frac{1}{N}\sum_{i=1}^{N}\sum_{b=1}^{24}|h_{\text{inv}}(\mathbf{X},\mathbf{x}^{(i)})_b - \mathbf{w}_b^{(i)}| + |h_{\text{fwd}}(\hat{\mathbf{X}},\hat{\mathbf{x}}^{(i)})_b - \mathbf{w}_b^{(i)}|.$$

We empirically set $\lambda_r = \lambda_s = 1$ and $\lambda_{skin} = 10$ throughout our experiments. For all models (single-view point cloud and full point cloud), we use the Adam [5] optimizer with a learning rate of $1e^{-4}$ and train on the training set (10 subjects of CAPE) for 200k iterations, with a mini-batch size of 12. The learning rate of $1e^{-4}$ is a common choice used in many previous works [8, 13, 15] and we did not specifically tune it.

## 2 Hyperparameters for MetaAvatar

In this section, we describe the hyperparameters used for our MetaAvatar model. We specify the hyperparameters for the two algorithms described in Section 4.1 and Section 4.2 of the main paper, respectively.

**Meta-learned Initialization of Static Neural SDFs (Sec. 4.1 of the main paper):** The algorithm is described as Alg. 1:

---

**Algorithm 1** Meta-learning SDF with Reptile [11]

---

    **Initialize**: meta-network parameters $\phi$, meta learning rate $\beta$, inner learning rate $\alpha$, max training iteration $N$, inner-loop iteration $m$, batch size $M$
1: **for** $i = 1, \ldots, N$ **do**
2:     Sample a batch of $M$ training samples $\{\hat{\mathbf{X}}^{(j)}\}_{j=1}^{M}$
3:     **for** $j = 1, \ldots, M$ **do**
4:         $\phi_0^{(j)} = \phi$
5:         **for** $k = 1, \ldots, m$ **do**
6:             $\phi_k^{(j)} = \phi_{k-1}^{(j)} - \alpha\nabla_{\phi}\mathcal{L}_{\text{IGR}}(f_{\phi}(\hat{\mathbf{X}}^{(j)})|_{\phi=\phi_{k-1}^{(j)}})$
7:         **end for**
8:     **end for**
9:     $\phi \leftarrow \phi + \beta\frac{1}{M}\sum_{j=1}^{M}(\phi_m^{(j)} - \phi)$
10: **end for**

---

where we set $\beta = 1e^{-5}$, $\alpha = 1e^{-4}$, $N = 160k$, $m = 24$ and $M = 3$. $m = 24$ is empirically set. The learning rates are found by starting at $1e^{-4}$ then decrease it by 10 times each time the model cannot converge. Same applies for hyperparameters of Alg. 2 and other ablation baselines. For both of the outer-loop update (line 9) and inner-loop update (line 6) we use the Adam [5] optimizer as we found that the SGD optimizer does not converge, probably due to the complex second-order gradients caused by the IGR [3] loss.

**Meta-learned Initialization of HyperNetwork for Dynamic Neural SDFs (Sec. 4.2) of the main paper:** The algorithm is described as Alg. 2. We set $\beta = 1e^{-6}$, $\alpha = 1e^{-6}$, and $m = 24$. Since the number of samples per inner-loop is random, we did not specify the maximum number of training iterations $N$, but rather train for 50 epochs. Again we use Adam for gradient updates of both the inner and the outer loop. For each MLP of the hypernetwork, we initialize the last linear layer with zero weights and biases, while all other linear layers are initialized using the default setting of PyTorch [12], *i.e.* the He initialization [4].

**Implementation Details:** For the IGR loss we use the same hyperparameters as in SIREN [16] for both algorithms. We normalize all query points in the canonical space to the unit cube $[-1, 1]^3$ according to the underlying SMPL registration of each frame. For depth rendering, we render the meshes into $250 \times 250$ depth images, which end up having roughly 3000-5000 points per frame; for both training and fine-tuning, we randomly duplicate (if the frame has less than 5000 points) or sample (if the frame has more than 5000 points) these points to make each frame contains exactly 5000 points such that we can process them in batches. For off-surface points, we randomly sample 5000 points in the unit cube $[-1, 1]^3$.

**Algorithm 2** Meta-learning hypernetwork with Modified Reptile

---

**Initialize**: meta-hypernetwork parameters $\psi$, pre-trained meta-SDF parameters $\phi^*$, meta learning rate $\beta$, inner learning rate $\alpha$, max training iteration $N$, inner-loop iteration $m$.

1: **for** $i = 1, \ldots, N$ **do**
2:      $\psi_0 = \psi$
3:      Randomly choose a subject/cloth-type combination $n$
4:      Uniformly sample $M \sim \{1, \ldots, D^{(n)}\}$ where $D^{(n)}$ is the number of datapoints of subject/cloth-type combination $n$
5:      Sample $M$ datapoints from subject/cloth-type combination $n$, denoting these datapoints as $\mathcal{S} = \{\{\mathbf{B}_b\}^{(j)}, \hat{\mathbf{X}}^{(j)}\}_{j=1}^{M}$
6:      **for** $k = 1, \ldots, m$ **do**
7:          $\mathcal{L} = \frac{1}{M} \sum_{(\{\mathbf{B}_b\}, \hat{\mathbf{X}}) \in \mathcal{S}} \mathcal{L}_{\text{IGR}}(f_\phi(\hat{\mathbf{X}})|_{\phi = g_{\psi_{k-1}}(\{\mathbf{B}_b\}) + \phi^*})$
8:          $\psi_k = \psi_{k-1} - \alpha \nabla_{\psi_{k-1}} \mathcal{L}$
9:      **end for**
10:     $\psi \leftarrow \psi + \beta(\psi_m - \psi_0)$
11: **end for**

---

During fine-tuning, we remove all points whose re-projection distance (Eq. (2)) is greater than 2cm.

Very occasionally (a dozen of frames out of more than 5000 frames), we observe visual artifacts caused by floating blobs which are the results of imperfect SDFs. We remove such floating blobs by keeping only the largest connected component of the extracted iso-surfaces. This post-processing does not impact the mesh-distance and normal consistency metrics as they consist only a very small portion of the results.

## 3 Hyperparameters for Baselines

We further describe the hyperparameters for baselines that we have compared to in Section 5 of the main paper.

### 3.1 NASA

We follow NASA [2] and train each model for 200k iterations while using the Adam optimizer [5] with the learning rate of $1e^{-4}$ and the batch size of 12. Each batch consists of 1024 uniformly sampled points and 1024 points sampled around the mesh surface; an additional set of 2048 points are sampled on the ground-truth mesh surface to compute the weight auxiliary loss.

### 3.2 LEAP

We follow the experimental setup presented in LEAP [9] and train it until convergence ($\approx$ 300k iterations) by using the Adam optimizer [5] (learning rate $1e^{-4}$) with the batch size of 30. Each batch consists of 2048 uniformly sampled points and 2048 points sampled around the mesh surface in the canonical space.

Since LEAP is designed for minimally-clothed human bodies, we extend it to learn more challenging clothing types. Specifically, we represent different clothing types via a one-hot encoding vector $\mathbb{R}^9$ and concatenate it to the LEAP's global feature vector.

### 3.3 SCANimate

We follow the architecture and hyperparameter settings as provided in the SCANimate [14] paper, and train the models in three stages: 80 epochs for pre-training the skinning nets, 200 epochs for training the skinning nets with the cycle-consistency loss, and 1000 epochs for training the neural SDF. For the ground truth clothed body data, we sample 8000 points on the clothed body meshes, and do so dynamically at each iteration.

Note that the full SCANimate pipeline is designed to work on raw scans, whereas in our settings, the clothed body meshes are registered meshes with the SMPL model topology (6890 vertices and 13776

**Cloth-deformation Evaluation**

Each HIT contains side-by-side comparison of two video clips, choose the clip (left or right) in which cloth-deformation looks more releastic

Use the buttons on the right of the screen to give your choice, and submit the HIT via the submit button. When the video fails to play, select "not playing".

Figure 1: User interface of our perceptual study

triangles). We strictly follow the full pipeline of SCANimate, which includes a step of removing distorted triangles (up to an edge stretching threshold of 2.0) after canonicalization. While this step works well for high-resolution raw scans, it may unnecessarily remove certain pose-dependent deformations when applying to the registered clothed body meshes in our settings. This happens most frequently around the armpit and elbow region, which, consequently, creates artifacts at these regions in the final posed body meshes. Such domain gap in the data can help explain the slightly compromised quantitative performance of SCANimate in Section 5.1 of the main paper. Nonetheless, SCANimate shows impressive qualitative results and perceptual scores, and yet our model still outperforms it in both metrics, demonstrating the effectiveness of our approach.

## 4   Hyperparameters for Ablation Baselines

In this section, we describe the hyperparameters for ablation baselines that we have compared to in Section 5.2 of the main paper. To recap, the ablation baselines include (1) a plain MLP that takes the concatenation of the relative joint-rotations (in the form of unit quaternions) and query points as input (MLP), (2) an MLP that takes the concatenation of the relative joint-rotations and the positional encodings of query point coordinates as input (PosEnc), and (3) a SIREN network that takes the concatenation of the relative joint-rotations and query points as input (SIREN).

**Shared Hyperparameters:** All ablation baselines (MLP, PosEnc and SIREN) are trained using Alg. 2 with $\phi^* = 0$. We set $\beta = 1e^{-5}$, $\alpha = 1e^{-4}$ and $m = 24$. Since these ablation baselines do not have an initialization like MetaAvatar does, we train them for 100 epochs instead of 50 epochs.

**PosEnc:** We use the 8th-order positional encoding [10] to encode the xyz-coordinates while leaving the pose inputs as it is.

**SIREN:** We initialize the weights of the network the same way as described in the original SIREN [16] paper.

## 5   Perceptual Study Details

The Amazon Mechanical Turk (AMT) user interface and instruction of our perceptual study are illustrated in Fig. 1. In each Human Intelligence Task (HIT), the user is provided with a side-by-side comparison video of MetaAvatar versus one baseline approach, the video corresponds to one action

| Per model fine-tuning time (hours) | | | | |
| --- | --- | --- | --- | --- |
| 00122-shortlong | 00122-shortshort | 00134-longlong | 00134-longshort | 00215-jerseyshort |
| 0.71 | 1.1 | 3.7 | 3.7 | 0.93 |
| 00215-longshort | 00215-poloshort | 00215-shortlong | 03375-blazerlong | 03375-longlong |
| 0.64 | 1.08 | 1.0 | 1.04 | 2.08 |

Table 1: **Per-model fine-tuning time with 100% data.** Our model runs with a fixed number of epochs, thus the fine-tuning time scales linearly with the amount of fine-tuning data. With 8 depth frames it only takes less than 2 minutes for fine-tuning.

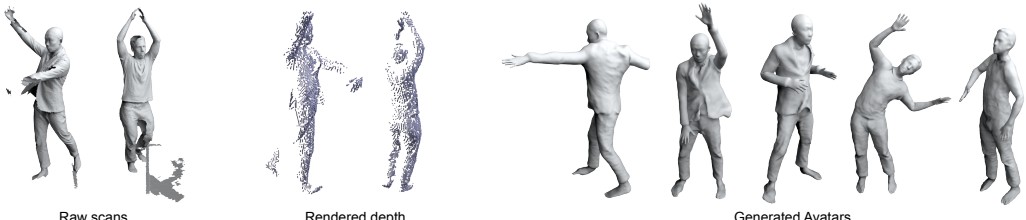

Raw scans      Rendered depth      Generated Avatars

Figure 2: Although our meta-model is learned on registered CAPE meshes, it generalizes well on depth images rendered from raw scans.

sequence of CAPE [7] which lasts 5-10 seconds. The left-right ordering is randomly shuffled for each video so that users do not become biased towards one side. To take internet/OS-related issues into account, we also provide a "Not playing" option in case certain users cannot play some videos on their devices. For each video, we ask 3 different users to independently choose the side on which they think the cloth-deformation looks more realistic. Given the total number of videos as $M$, the perceptual score is calculated as $PS = \frac{P}{3M-N}$, where $N$ is the number of user choices that choose "Not playing", and $P$ is the number of user choices that choose the baseline approach over MetaAvatar. Thus the perceptual score is in the $[0, 1]$ range, a score that is less than $0.5$ means the baseline looks less realistic than MetaAvatar, while a score that is greater than $0.5$ indicates that the baseline looks more realistic than MetaAvatar. For MetaAvatar, we manually define its perceptual score to be $0.5$.

For salary, we provide a 0.02$ reward for each HIT. For evaluation against baselines, we evaluated a total number of 1143 HITs, which costs 34.29$ (22.86$ for reward and 11.43$ for AMT). For evaluation of few-shot learning, we evaluated a total number of 1905 HITs, which costs 57.15$ (38.10$ for reward and 19.05$ for AMT).

## 6   Timing Details

We report the time required for fine-tuning on the full amount of data for each subject/cloth-type combination in Tab. 1. Timings are measured on a single NVIDIA V100 GPU of our internal cluster. We fine-tune 256 epochs for each model. In comparison, NASA runs at a fixed number of iterations (200k) which usually takes more than 10 hours. SCANimate needs to train subject/cloth-type-specific skinning networks and neural SDFs which takes more than 10 hours and can be as much as 30 hours; the neural SDF training alone takes 1000 epochs per model. LEAP takes two days to train on all training and fine-tuning data.

## 7   Additional Qualitative Results

In this section we show additional qualitative results, including 1) results on depth rendered from raw scans, 2) results on real depth images, 3) results on single raw scan, 4) comparison with pre-trained SCANimate model. All qualitative results are in video format and can be found in our project page: https://neuralbodies.github.io/metavatar/

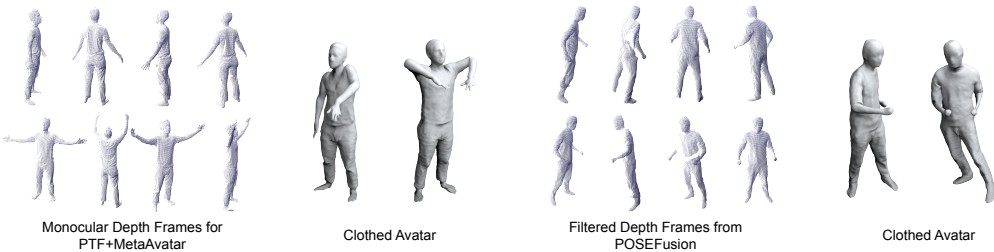

Monocular Depth Frames for PTF+MetaAvatar    Clothed Avatar    Filtered Depth Frames from POSEFusion    Clothed Avatar

Figure 3: **Left:** inputs to PTF and MetaAvatar and the generated avatar. **Right:** filtered and fused depth points from POSEFusion pipeline and the generated avatar.

## 7.1 Depth Rendered from Raw Scans

To demonstrate that our meta-learned is robust to domain gaps of data, we render depth images from the raw scans of the CAPE [7] dataset and demonstrate that our meta-learned model, which was trained on registered meshes, generalizes well on raw scans. Note that raw scans often come with large holes, noisy surfaces, redundant background points, and missing parts (*e.g.* left side of Fig. 2). To remove the background points such as floors and walls, we use the ground-truth CAPE meshes as a reference, removing any raw scan point whose distance is greater than 2cm from the corresponding ground truth clothed CAPE mesh. To handle the missing hands/feet problem, we replace them with hands and feet of the underlying SMPL registrations. We fine-tune our model on the 00122 subject with shortlong cloth-type and the 03375 subject with blazerlong cloth-type since we only have access to scans of these two subjects; for subject 00122 we fine-tuned on one action, while for 03375 we follow the same protocol which fine-tunes the model on trial1 sequences and test on trial2 sequences. We show several static frames in Fig. 2.

To further demonstrate the robustness of our approach to noisy SMPL registrations, we also utilize a recently released work PTF [17], which is able to register SMPL to single-view point clouds, to register SMPL to 8 frames of single-view point clouds rendered from CAPE raw scans. Using these noisy registrations and rendered depth frames, we fine-tuned an avatar and show the pose extrapolation results in Fig. 3.

## 7.2 Real Depth Images

Real depth sensors often give noisy outputs, and it is necessary to use tracking and fusion techniques to filter out noise and outliers. We thus utilize POSEFusion [6] to obtain the necessary data for creating avatars. The input to POSEFusion is a monocular RGBD video of a clothed person moving and rotating, showing both his/her frontal and back views. It uses tracking and fusion, guided by SMPL estimations, to fuse invisible parts from future frames to current frames, such that it can reconstruct the full-body mesh at each RGBD frame. Given the reconstructed full-body meshes as well as SMPL registrations, we render the first 8 frames of reconstructions and use these 8 frames along with their estimated SMPL fits to create our avatar. The avatar is then animated with estimated SMPL registrations of the rest of the sequence ( 210 frames) and sample poses from CAPE dataset ( 140 frames). Sample results are illustrated in Fig. 3.

## 7.3 Single Scan Animation

To additionally demonstrate the robustness of our meta-learned model, we also fine-tuned an avatar using a single full-body scan. Sample results are showed in Fig. 4. Note that our model is meta-learned on rendered depth images yet it also yields reasonable results on full-body scans.

## 7.4 Comparison with Pre-trained SCANimate Model

To demonstrate the advantage of meta-learning, we also compare our results with results fine-tuned from pre-trained SCANimate model. We use the official SCANimate release code, which comes with 16 training raw scans of subject 03375-shortlong and several pre-trained models on different subject/cloth-type combinations. We found that 00096-shirtlong is in our training set and the model

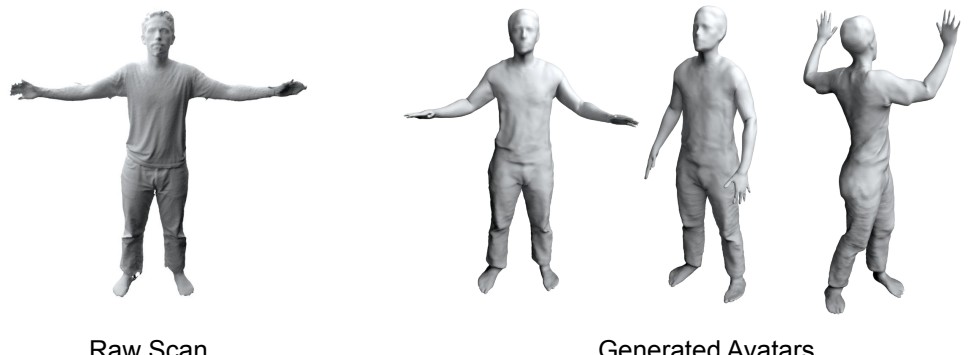

Raw Scan              Generated Avatars

Figure 4: **Single Scan Animation**. Our meta-learned model is able to animate a single scan.

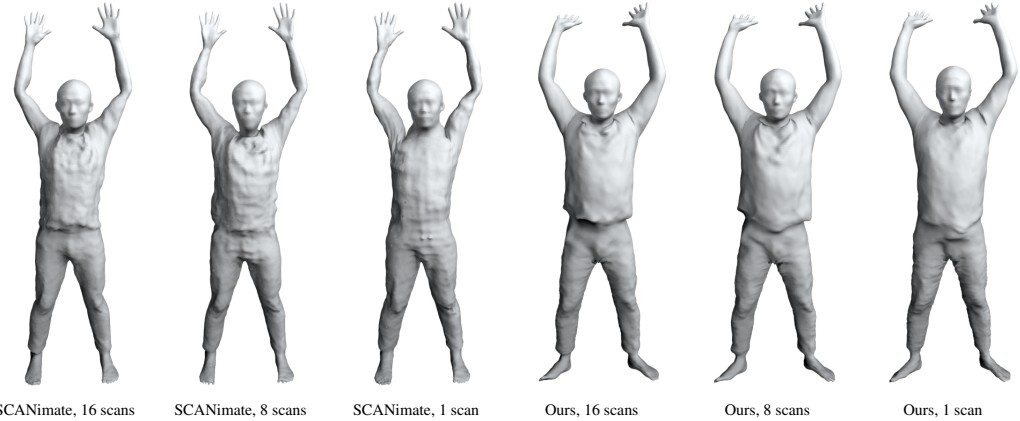

SCANimate, 16 scans  SCANimate, 8 scans  SCANimate, 1 scan  Ours, 16 scans  Ours, 8 scans  Ours, 1 scan

Figure 5: **Comparison with SCANimate on limited data**. Note that SCANimate sets hands and feet poses to 0. With limited data, SCANimate cannot model cloth deformation properly and has obvious artifacts around elbows. In comparison, our model can produce reasonable cloth deformations and body shapes with limited data.

has similar body shape and cloth-type to 03375-shortlong. We thus fine-tune the pre-trained model of 00096-shirtlong with provided raw scans of subject 03375-shortlong, using the default configuration of SCANimate.

We fine-tune SCANimate with 16/8/1 raw scan(s) to verify its performance on reduced data and compare it to our model. We animate the fine-tuned avatars with 03375-shortlong's trial2 action sequences; these actions have not been seen in either training data or fine-tuning data. Sample results are shown in 5.

## 8   Limitations

As illustrated in Fig. 6, the proposed MetaAvatar can generalize to the blazer outfit which is unseen during meta-learning. However, the model needs to see a moderate amount of data in order to represent the cloth dynamics of this outfit, otherwise it cannot capture the specific dynamics such as the tails of the blazer.

Another limitation is that MetaAvatar (and all other baselines) relies on accurate SMPL registrations to input point clouds. In the future, we aim to combine our work with the state-of-the-art parametric human body registration approaches [1, 17] to enable the joint learning and optimization of neural avatars and SMPL registrations.

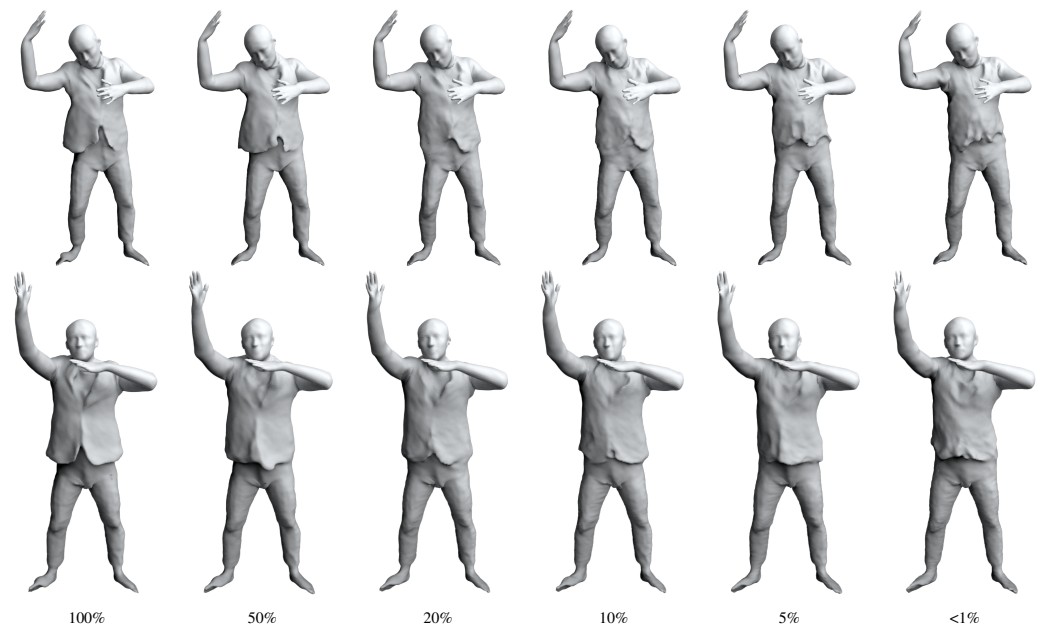

| 100% | 50% | 20% | 10% | 5% | <1% |

Figure 6: **Limitation**. For the challenging blazer outfit which is not seen during meta-learning, our model fails to capture its dynamics with limited fine-tuning data.