# OpenReview forum: "MetaAvatar: Learning Animatable Clothed Human Models from Few Depth Images"
_NeurIPS.cc/2021/Conference — NeurIPS 2021 Poster_

### Official Review · Reviewer_xRBs · 2021-07-15

**Rating:** 6
**Confidence:** 2

**Summary:**

This paper proposes a novel learning method to generate controllable neural SDF for non-rigid clothed human body from a few depth scans. The entire pipeline consists of three main parts, i.e., canonicalization, neural SDF, and hyper-net. The neural SDF takes canonicalized points as inputs and learns a static representation. The hyper-net is used to fine-tune the static neural SDF and generate dynamic SDF efficiently for forward LBS based animation.


**Limitations And Societal Impact:**

Please see the main review part.

**Main Review:**

As far as I am concerned, the method proposed in this work is novel and technically sound. The target application can be useful for animation of clothed 3D human body from a few depth incomplete input. The presented animation results demonstrate some advantages over those obtained via state-of-the-art methods in certain cases from the CAPE dataset.

On the other hand, I think this paper needs some major revisions on paper writing, especially Section 4. I suggest to add an overview section or paragraph to present the entire pipeline and clearly describe the input and output of each stage/component. The inference part should be put into an individual subsection. In Section 5, the CAPE dataset contains 15 human objects but only 14 are used for the experiments (among which 10 for training, 4 for validation and test). Is there any particular reason for this choice? Also, the depth frames are synthetically rendered from the CAPE dataset. There are no examples using input from real depth sensor or dataset other than CAPE.

Overall, I think this work is technically sound but the presented results are not convincing enough.

**Time Spent Reviewing:**

6

---

> ### Author Response · Authors · 2021-08-09
> **Clarification for Datasets and Additional Qualitative Results**
>
> Thank you for your acknowledgement for the novelty of our work and the constructive feedback! We will address your questions and concerns in the following:
>
> Concern 1: Paper writing needs revision
>
> As suggested, in the final version of our paper, we will 1) add an overview paragraph introducing training and inference pipelines, before diving into details 2) make inference pipeline a separate subsection.
>
> Concern 2: Why only 14 subjects from CAPE are used?
>
> We ignore subject 00159 as it has significantly less data compared to other subjects (please see footnote of page 7). Specifically, it only has 3 action sequences for each cloth-type, while other fine-tuning/validation subjects have at least 8 sequences for each cloth-type.
>
> Concern 3: No examples for real depth sensors
>
> Please see our response to the first concern of reviewer x2Rq. In short, we include additional qualitative videos at this anonymous dropbox link:
>
> https://www.dropbox.com/sh/9xavvkwaf6qrkuf/AADE86SVtcaqqC_ZUckeq9Ema?dl=0
>
> For examples of monocular Kinect, you can refer to video (2).
>
> Concern 4: No additional dataset other than CAPE was used
>
> To our best knowledge, CAPE is the only public dataset that provides meshes of clothed humans along with their SMPL registrations. Without full-body meshes and good SMPL registrations, it is impossible for us to compare to the baselines in the main paper, as they all require full-body meshes and SMPL registrations as inputs.

---

### Official Review · Reviewer_x2Rq · 2021-07-16

**Rating:** 8
**Confidence:** 4

**Summary:**

This paper presents a framework that can learn an animatable implicit avatar for clothed humans from very few observations. The key idea is using meta-learning to learn a prior (specifically speaking, a good initialization) to enable fast fine-tuning at inference time. To enable animations of various cloth types, this paper proposes to perform meta-learning on a hyper-network which controls the parameters of the neural implicit function conditioned on pose configurations. Results show that the proposed method is able to generate clothed human avatars from few shot observations.


**Limitations And Societal Impact:**

The authors have discussed the limitations in the supplemental document and the potential societal impact in Section 6. No more improvement is needed.



**Main Review:**

### Strengths:
* The idea of meta-learning a pose-conditioned hyper-network is interesting and inspiring. To the best of my knowledge, this paper is the first one that applies meta-learning in human modeling. The usage of meta-learning can accelerate network optimization by a large margin, e.g., from 10 hours to 1.6 hours as presented in the paper. The design of meta-learning a hyper-network is also a valuable idea for the community.

* This paper is well motivated and addresses an important problem: current methods for neural SDFs-based clothed avatars take too much optimization time. Compared to SOTA methods like SCANimate, the proposed method can generate comparable and even better results, while being much more efficient.

* This paper is overall well-written and easy to follow.

### Weaknesses:
* My major concern is about the claim that this method requires "only" depth frames as input.  Although the authors keep emphasizing that their method only needs partial depth observations for fine-tuning, in fact the method assumes perfect SMPL registrations and ground-truth bone transformations, which is only possible when complete 3D scans are available. In the experiments, the authors only use depth rendering from *complete* CAPE meshes. In the supplemental material, the authors conduct an additional experiment on raw scans, but that experiment still requires ground-truth SMPL registrations to clean and complete the scans, and also to calculate the bone transformations. In one word, I don't think the experiments can support the claim in L55-56 ("given only a few depth observations of an unseen clothed human as inputs") because of its dependence on ground-truth SMPL registrations. It would make this paper stronger if the authors can conduct experiments on *real* partial depth inputs, i.e., raw depth images captured by an RGBD sensor like Microsoft Kinect or Intel RealSense.

* The numerical evaluation in Table 2 is not very convincing. The numbers in the last 3 columns are really closed to each other. Table 2 makes me feel that the performance gains mainly stem from the usage of SIREN activations, and that the slight improvement in the last column is due to the increased number of network parameters.


### Minor:
* L226: Although reader can guess that the meshes are rotated around the yaw axis, it would be better if the authors can explicitly specify it.

* L311: to representation -> to represente


**Time Spent Reviewing:**

6 hours.

---

> ### Author Response · Authors · 2021-08-09
> **We Include Addtional Qualitative Results to Address Your Concerns**
>
> Thank you for your acknowledgement for the novelty of our work and the constructive feedback! To address your concerns, we include additional qualitative results, they can be found in this anonymous dropbox link:
>
> https://www.dropbox.com/sh/9xavvkwaf6qrkuf/AADE86SVtcaqqC_ZUckeq9Ema?dl=0
>
> There are 4 qualitative videos: (1) an avatar created from SMPL fits to rendered depth images (2) an avatar created with fused and filtered depth images and SMPL fits to real monocular RGBD video (3) an avatar created from a single full-body scan (4) a qualitative side-by-side comparison between meta-hypernetwork and meta-SIREN. In the following we will address your concerns by referring to these qualitative videos.
>
> Concern 1: The method depends on ground-truth SMPL, and getting perfect SMPL fittings is only possible on full-body scan; demo of real RGBD sensors would be desirable.
>
> We address this concern by video (1), (2) and (3)
>
> Video (1): We utilize a recently released work, PTF [1], which is able to register SMPL to single-view point clouds. We register SMPL to 8 frames of single-view point clouds rendered from CAPE raw scans. Using these registrations and rendered depth frames, we fine-tuned an avatar and show the pose extrapolation results.
>
> Video (2): We found that due to domain gaps between rendered depth and real depth from RGBD sensors, PTF does not work well on real RGBD inputs when trained on rendered depth. Real RGBD sensors often give noisy outputs, and it is necessary to use tracking and fusion techniques to filter out noise and outliers. We thus utilize POSEFusion [2] to obtain the necessary data for creating avatars. The input to POSEFusion is a monocular RGBD video of a clothed person moving and rotating, showing both his/her frontal and back views. It uses tracking and fusion, guided by SMPL estimations, to fuse invisible parts from future frames to current frames, such that it can reconstruct the full-body mesh at each RGBD frame. Given the reconstructed full-body meshes as well as SMPL registrations, we render the first 8 frames of reconstructions in the same way as we did in our paper and use these 8 frames along with their estimated SMPL fits to create our avatar. The avatar is then animated with estimated SMPL registrations of the rest of the sequence (\~210 frames) and sample poses from CAPE dataset (\~140 frames).
>
> Video (3): To additionally demonstrate the robustness of our meta-learned model, we also fine-tuned an avatar using a single full-body scan. Note that our model is meta-learned on rendered depth images yet it also yields reasonable results on full-body scans.
>
> Overall, given monocular depth images, existing works can estimate SMPL parameters, e.g. PTF [1]  for the monocular depth frames rendered from raw scans and POSEFusion [2] for the monocular noisy depth frames from Kinect.  As shown in the newly provided videos, given the estimated SMPL parameters, our meta-learned model is able to produce convincing  animatable avatars. Furthermore, our meta-learned model is versatile enough to generate avatars from few depth images as well as a single full-body scan (i.e. single-scan animation), both of which are not possible for existing works (e.g. SCANimate, LEAP, NASA).
>
> Concern 2: In Table 2, improvement of using meta-learned hypernetwork is small compared to meta-learned SIREN networks.
>
> We argue that distance metrics are proxy measurements and they do not always reflect qualitative improvements. This is most likely due to the stochastic nature of cloth deformations. Thus, we present additional qualitative comparison between meta-learned hypernetwork and meta-learned SIREN networks in video (4); overall, meta-hypernetwork can produce significantly better clothing details.
>
> [1] Wang et al. Locally Aware Piecewise Transformation Fields for 3D Human Mesh Registration, CVPR 2021
>
> [2] Li et al. POSEFusion: Pose-guided Selective Fusion for Single-view Human Volumetric Capture, CVPR 2021

---

### Official Review · Reviewer_XKUG · 2021-07-16

**Rating:** 8
**Confidence:** 5

**Summary:**

This paper aims to create generalizable and controllable neural signed distance fields from monocular depth observations. The authors propose a hypernetwork to represent neural SDFs across different human shapes and cloth types. For fast fine-tuning, they additionally leverage meta-learning to learn an initialization of the hypernetwork. This method achieves impressive reconstruction results and demonstrates a cool application that reconstruct high-quality human shapes from 8 monocular depth frames in 2 minutes.

**Ethics Review Area:**

["I don’t know"]

**Limitations And Societal Impact:**

I do not see potential negative societal impact of their work.

**Main Review:**

Originality:

1. This paper propose a novel representation for human shapes. It first learns a meta-SDF to fit canonical 3D subjects with diffrent genders, body shapes, cloth types, and poses. To model the shape deformation caused by the poses, it then meta-learn a hypernetwork that takes the poses as input and predicts the residuals to the parameters of the previously learned meta-SDF.
2. This representation can be seen a combination of coarse and fine representations, which effectively and efficiently models the shape deformation.

Quality:

1. The submission is technically sound.
2. The claims are well supported.
3. The comparisons and ablation studies are sufficient.

Clarity:

1. The submission is clearly written and well organized.

Significance:

1. This paper proposes a method that quickly reconstructs high-quality human shapes from monocular depth frames, which is very practical and largely promotes the field of human reconstruction. It will probably enlighten follow-up works.

**Time Spent Reviewing:**

2 hours

---

> ### Author Response · Authors · 2021-08-10
> **Thanks!**
>
> We highly appreciate your acknowledgement for our work!

---

> > ### Comment · Reviewer_XKUG · 2021-08-19
> > **One additional question about meta learning**
> >
> > When I read this paper again, I have one additional question about the meta learning. In Section 5.3, the authors only compare the performances and training time on their model, I am curious about the performances and training time of other model, like SCANimate, if the amount of fine-tuning data is reduced.
> >
> > I know that this is unfair since SCANimate is not pre-trained on other person data. So here I am curious about the performance of SCANimate if it is first trained on one of 10 subjects in the training set and then fine-tuned on one unseen subject.
> >
> > I think that this comparison is important, since the main contribution of this paper is proposing a meta learning algorithm to reduce the amount of training data and training time.

---

> > > ### Author Response · Authors · 2021-08-23
> > > **We Include Additional Qualitative Results to Address Your Question**
> > >
> > > Thank you for your suggestion! We followed your suggestion and added additional experiments as requested:
> > >
> > > **Setting**:  We use the official SCANimate release code, which comes with 16 training raw scans of subject 03375-shortlong and several pre-trained models on different subject/cloth-type combinations. We found that 00096-shirtlong is in our training set and the model has similar body shape and cloth-type to 03375-shortlong. We thus fine-tune the pre-trained model of 00096-shirtlong with 16 raw scans of subject 03375-shortlong, using the default configuration of SCANimate.
> > >
> > > We fine-tune SCANimate with 16/8/1 raw scan(s) to verify its performance on reduced data and compare it to our model. We animate the fine-tuned avatars with 03375-shortlong’s trial2 action sequences; these actions have not been seen in either training data or fine-tuning data.
> > >
> > > **Results**: The result videos can be found at this anonymous dropbox link:
> > >
> > > https://www.dropbox.com/sh/mh1ybqehjbmz6hy/AADrSBX3hzr83rXyhKyqHHewa?dl=0
> > >
> > > In each of the videos, we visualize the results of both our model and SCANimate at 24/128/256/1024 fine-tuning epochs. We summarize our advantages over SCANimate in the following:
> > >
> > > *(1) Robustness for hard poses*: with 16 raw scans, SCANimate already has many failure cases at hard poses (e.g. squatting, ballet dancing, running) while our model can handle such poses properly. With a reduced amount of data (e.g. 8 scans and 1 scan), the results on hard poses of SCANimate only get worse while our model can still handle those poses properly.
> > >
> > > *(2) Robustness for reduced data*: as one can see, with 8 scans SCANimate starts to have consistent artifacts such as shrinking arms/neck and collapsed body. This only gets worse with a single scan. On the other hand, our model does not produce such artifacts and it can produce reasonable body shape and cloth deformation with 8 scans or even a single scan.
> > >
> > > *(3) Fast convergence*: our model can achieve good cloth deformation details with only 24 epochs. With the same amount of fine-tuning epochs SCANimate hardly recovers any cloth deformation. Note that to fine-tune SCANimate, we use the default learning rate of its official code, which is 4e-3, while our meta-learned model is fine-tuned with a learning rate of 1e-6. Thus the fast convergence of our model should be attributed to our model initialization instead of learning rate settings. With 16 scans, fine-tuning SCANimate for 24/128/256/1024 epochs takes about 0.7/4/7/30 minute(s), while fine-tuning our model for 24/128/256/1024 epochs take about 0.23/1/2/9 minute(s); the timing is of course a coarse estimation as it is affected by many factors such as CPU/disk work-load of the machine, data-loader efficiency and network size, and both SCANimate and our model have room for optimization, but the important observation is that in terms of few-shot learning, our model can achieve better qualitative results than SCANimate with as few as 24 fine-tuning epochs.
> > >
> > > Please let us know if you have more questions or concerns, thanks!

---

> > > > ### Comment · Reviewer_XKUG · 2021-08-24
> > > > **Response**
> > > >
> > > > Thanks for your responses!
> > > > I think that the experiments makes the submission stronger.

---

### Official Review · Reviewer_ZigT · 2021-07-17

**Rating:** 7
**Confidence:** 4

**Summary:**

This paper looks at learning animatable avatars given a small number of depth images with associated known SMPL pose transformations. The proposed approach (similar to prior work e.g. SCANimate) learns: a) forward/inverse skinning weight predictors,  which given a query point x in canonical/posed space respectively predict the forward/inverse LBS weights, and b) a canonical space implicit SDF function, which conditioned on SMPL pose, predicts a pose-dependent shape s.t. its forward transformation looks correct.

While the paper uses a fixed network for a) above, the key contributions relate to how the pose-dependent canonical shape is modeled. This paper proposes to capture each object via a hyper-network which predicts the weights of the SDF function conditioned on SMPL pose parameters. It shows that such a hypernetwork can be meta-learned using training data with strong supervision, but can then be finetuned using only a small number of depth images in varying poses for a novel instance.

The experiments convincingly show the benefits of this approach across different datasets and in particular show how using only a small number of depth images suffice.


**Limitations And Societal Impact:**

Yes

**Main Review:**

This is a very well-motivated and a very-well executed paper which looks at learning an avatar for a novel instance given a small number of depth images with known poses. In particular:

a) I really liked the insight of modeling the pose-dependent shape variations of each instance via a  hyper-network, and the paper convincingly demonstrates that this network can be effectively finetuned via a small number of depth images (unlike prior methods which use full scans/meshes).

b) The insight of factoring this hyper-network into a base network, only whose residual weights are predicted is also an elegant one.

c) The approach for predicting generalizable skinning weights via a convolutional network is also well-motivated, although I am not sure why a similar hyper-network based approach would not be suitable (especially for forward skinning weights).

d) The experiments, and the visualizations in the supplementary convincingly show that this approach allows more detailed reconstruction over the baselines - similar to prior methods which overfit to a single instance, it can capture details of the 3D shape, while only requiring a fraction of data or optimization time.

e) The ablations, in particular the MLP, PosEnc, and SIREN baselines clearly justify the benefits of learning a hypernetwork to model each instance instead of a 'normal' conditional network to predict shape by concatenating the SMPL pose.

Overall, this is a good paper that proposes a novel and insightful solution for learning animatable avatars from a small number of depth images, and I would argue for acceptance.

**Time Spent Reviewing:**

3

---

> ### Author Response · Authors · 2021-08-09
> **Clearification for Skinning Networks**
>
> Thank you for your time and acknowledgement for our work!
>
> Regarding the skinning networks, we found these simple convolutional skinning networks work very well already. On the other hand, forward skinning weights prediction is a relatively easy problem, as the inputs are always in canonical pose and thus have very little variance. We thus did not explore other alternatives for skinning networks.

---

### Decision · Program_Chairs · 2021-09-27

**Decision:**

Accept (Poster)

**Comment:**

This submission introduces a method that enables the generation of realistic clothed human avatars from monocular depth images. While the initial reviews are mixed, after rebuttal, all reviewers are positive and recommend acceptance.  The AC agrees.  The authors should try to address the reviewers' concerns in the camera-ready version. This includes adding the results from the rebuttal period, clarifying the concerns on meta-learning as reviewer XKUG suggested, among others.